# A structural dynamics model for how CPEB3 binding to SUMO2 can regulate translational control in dendritic spines

Xinyu Gu[1,2], Nicholas P. Schafer[1,2], Carlos Bueno[1], Wei Lu[1], Peter G. Wolynes[1,2]*

1 Center for Theoretical Biological Physics, Rice University, Houston, Texas, United States of America,
2 Department of Chemistry, Rice University, Houston, Texas, United States of America

* pwolynes@rice.edu

**Data Availability Statement:** Original data and codes are available at https://doi.org/10.5281/zenodo.5677353.

## Abstract

A prion-like RNA-binding protein, CPEB3, can regulate local translation in dendritic spines. CPEB3 monomers repress translation, whereas CPEB3 aggregates activate translation of its target mRNAs. However, the CPEB3 aggregates, as long-lasting prions, may raise the problem of unregulated translational activation. Here, we propose a computational model of the complex structure between CPEB3 RNA-binding domain (CPEB3-RBD) and small ubiquitin-like modifier protein 2 (SUMO2). Free energy calculations suggest that the allosteric effect of CPEB3-RBD/SUMO2 interaction can amplify the RNA-binding affinity of CPEB3. Combining with previous experimental observations on the SUMOylation mode of CPEB3, this model suggests an equilibrium shift of mRNA from binding to deSUMOylated CPEB3 aggregates to binding to SUMOylated CPEB3 monomers in basal synapses. This work shows how a burst of local translation in synapses can be silenced following a stimulation pulse, and explores the CPEB3/SUMO2 interplay underlying the structural change of synapses and the formation of long-term memories.

## Author summary

Local translation of specific synaptic proteins provides the molecular basis for the structural change in dendritic spines, which is essential for long-term memories. A functional prion-like RNA-binding protein, CPEB3, has been proposed as a synaptic tag to regulate local translation in dendritic spines. More interestingly, the soluble CPEB3 monomers repress translation, whereas the CPEB3 aggregates activate the translation of its target mRNAs. The CPEB3 aggregates, however, that act as long-lasting prions providing "conformational memory", may raise the problem of translational activation being unregulated. Here, we propose a computational model of the complex structure between CPEB3 RNA-binding domain (CPEB3-RBD) and small ubiquitin-like modifier protein 2 (SUMO2). Free energy calculations suggest that the allosteric binding of CPEB3 with SUMO2 can confine the CPEB3-RBD to a conformation that favors RNA-binding, and thereby can amplify its RNA-binding affinity. Combining this model with previous experiments showing that CPEB3 monomers are SUMOylated in basal synapses but become

**Funding:** XG, NPS, CB, WL, and PGW were supported by the NSF Division of Chemistry RAISE grant CHE-1743392 and by the Center for Theoretical Biological Physics, sponsored by the NSF Division of Physics grant PHY-2019745. CB was also supported by the PoLS Student Research Network sponsored by the NSF Division of Physics grant 1522550. PGW was supported by the D. R. Bullard-Welch Chair at Rice University, Grant C-0016. The funders had no role in study design, data collection and analysis, decision to publish, or preparation of the manuscript.

**Competing interests:** The authors have declared that no competing interests exist.

deSUMOylated and start to aggregate upon stimulation, we suggest a way in which the translational control of CPEB3 can be switched back to a repressive mode after a stimulation pulse, through an RNA binding shift from binding to CPEB3 fibers to binding to SUMOylated CPEB3 monomers in basal synapses.

## 1 Introduction

The remodeling of the actin cytoskeleton in dendritic spines serves as the molecular basis for the structural changes of synapses [1] which are associated with the formation of long-term memories [2]. A synaptic tag is required to label budding synapses so as to regulate the local translation of actin mRNA and the mRNA of other synaptic proteins over long periods of time. A promising candidate for the synaptic tag is the mammalian cytoplasmic polyadenylation element-binding protein 3 (CPEB3). Mammalian CPEB3 [3] and its homologs, ApCPEB in Aplysia [4] and Orb2 in Drosophila [5, 6], have been shown to regulate the translation of synaptic proteins by binding with the U-rich CPE sequence in the 3' untranslated region (UTR) of target mRNAs. Targets of CPEB3 include the message RNAs for actin [7] and other protein components essential for long-term synaptic persistence, like GluA1 and GluA2 [3]. The longevity of the synaptic tag is traced to the fact that, CPEB3 can form self-sustaining prion-like aggregates [8] that resist rapid molecular turnover [9]. Understanding the system dynamics of the CPEB3-actin regulation network [10] is essential to see how CPEB3 is able to consolidate synaptic structure and facilitate the formation of long-term memories.

CPEB3 in dendritic spines can be found either in a monomeric state or in an amyloid-like aggregated state. Intriguingly, the monomeric CPEB3 has been found to repress translation, while the aggregated form of CPEB3 activates the translation of its target mRNAs [11–13]. One idea that has been proposed is that monomeric CPEB3 localizes its target mRNAs through forming gel-like processing bodies (P bodies) in which translation is repressed and that CPEB3 aggregation into a prion form simply releases those mRNAs [13]. A structural study of a CPEB3 homolog, Orb2, has found, however, that Orb2 aggregated in the prion form still binds target mRNAs and interacts with various proteins that might further recruit translation promoting factors, like polyadenylation complex [14]. We have recently suggested that changes in the activation and repression of translation ability by CPEB3 can be explained by a vectorial channeling effect in which the recycling of ribosomes depends on the structure of CPEB3/RNA assemblies. [15]. Vectorial channeling arises from the vectorial nature of mRNA translation, along with the structurally polarized nature of the mRNA/prion assembly. This structural synergy allows CPEB3 aggregates to form a local translation factory assembly lines in which ribosomes are more efficiently recycled than they are by the monomeric form and thus turn on the translation of dormant target mRNAs. The CPEB3 aggregates, which function then as synaptic tags providing "conformational memory", would be stable in synapses. Such stability, by itself would seemingly lead to continuous activation of local translation if aggregates were always found bound with their target mRNAs. This raises the question of whether and how such translational enhancement could be turned off so that synapses may return to a new basal state. (As illustrated in Fig 1).

One possibility that has been suggested involves the SUMOylation of CPEB3. There is evidence that SUMOylation, a reversible post-translational modification, can regulate CPEB3 function. SUMO proteins, small ubiquitin-like modifier proteins, can be covalently attached to lysine residues of target proteins. After such SUMOylation, the SUMO modification can subsequently be deconjugated (deSUMOylation). Monomeric CPEB3 is found to be SUMOylated

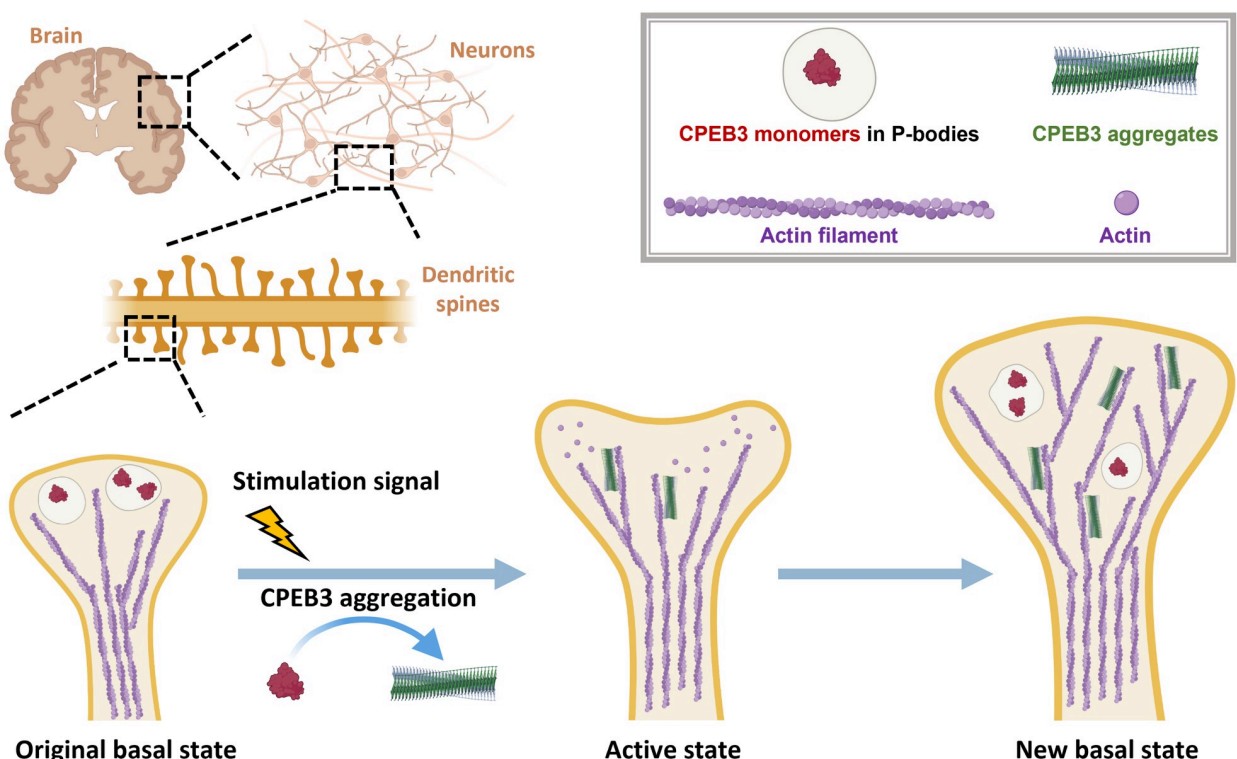

**Fig 1. The conformational changes of CPEB3 provide a mechanism for the switchable translational control in dendritic spines.** In a basal state, monomeric CPEB3 (shown in red) is colocalized with P-bodies and represses the translation of its target mRNAs. In response to a stimulation signal, CPEB3 monomers are released from P-bodies and then aggregate into CPEB3 fibers (shown in green). CPEB3 fibers (using the vectorial channeling mechanism) activate the local translation of synaptic proteins, including the actin proteins which are the molecular basis for the growth of the spines. The stability of the remaining prion-like CPEB3 aggregates would raise a problem of how the translational activation is turned down after a stimulation pulse. SUMO binding provides a route to return to the new basal state. A legend is shown at the upper-right corner.

by SUMO2, one homolog of the SUMO protein family, and in that form it is soluble in basal synapses. After stimulation, CPEB3 becomes deSUMOylated and aggregates. [12] SUMOylation of CPEB3 has been found to facilitate its colocalization to P bodies and is crucial for repressing translation. [13] DeSUMOylation of CPEB3 exposes the prion-like domain (PRD) and the actin-binding domain (ABD) of CPEB3 and triggers actin-facilitated CPEB3 aggregation as a result. [16] An interesting fact in thinking about the system biology of SUMO2/CPEB3 is that SUMO2 mRNA itself is one of CPEB3's targets. Therefore, there is a negative feedback loop between CPEB3 aggregation and SUMO2 synthesis [12]: The CPEB3 prion activates the translation of SUMO2 which can then be used for CPEB3 SUMOylation and lead to a return to translational repression. This negative feedback loop could thereby serve as a control on the balance between activation and repression of translation by CPEB3.

In this paper, we put forward a structural dynamics model of the interaction between SUMO2 and the RNA-binding domain (RBD) of SUMOylated CPEB3. We developed a structural model for the SUMO2/CPEB3-RBD complex through computational modeling using the Associative memory, Water-Mediated, Structure and Energy Model (AWSEM) [17, 18], a coarse-grained protein force field which has been optimized using energy landscape theory [19]. The AWSEM software has proved quite successful in predicting both monomeric protein structures and the structures of protein complexes. [17, 20] The SUMO2 protein simultaneously interacts with two distinct surfaces of the CPEB3 RNA-binding domain. By doing so, it closes the conformation of the RNA binding domain into a structure favorable for RNA

binding. Using the AWSEM force field and combining it with the Three Sites Per Nucleotide model 2 (3SPN2), a coarse-grained force field for nucleic acids developed by the de Pablo group [21, 22], we have calculated the free energy profile for RNA dissociation from the SUMO2/CPEB3-RBD complex. These calculations show that the RNA-binding free energy for the SUMO2/CPEB3-RBD complex is 2 kcal/mol larger than that for isolated RBD in deSU-MOylated CPEB3. We propose that the resulting difference in the RNA-binding affinity between the two forms causes a shift in the equilibrium of RNA binding from binding to the deSUMOylated CPEB3 aggregates to binding to SUMOylated CPEB3 monomers when synapses return to a new basal state. In this way, the translational control of CPEB3 becomes switchable in response to input signals: After stimulation, CPEB3 is deSUMOylated, so that CPEB3 aggregates and thereby activates translation of SUMO2 and other synaptic proteins. Monomeric CPEB3 once SUMOylated with newly synthesized SUMO2 proteins shifts the RNA-binding equilibrium, so that target mRNAs are recruited into P bodies by the SUMOylated CPEB3 and thereby are silenced.

## 2 Results

### 2.1 A full-length CPEB3-RBD model details its inter-domain interactions

The CPEB family consists of four isoforms in mammals, CPEB1–4. The composition of the C-terminal RBD in all four isoforms is highly conserved. The RBD includes two RNA recognition motifs (RRMs) and one zinc-finger domain (ZnF) as shown in Fig 2A. These isoforms can be

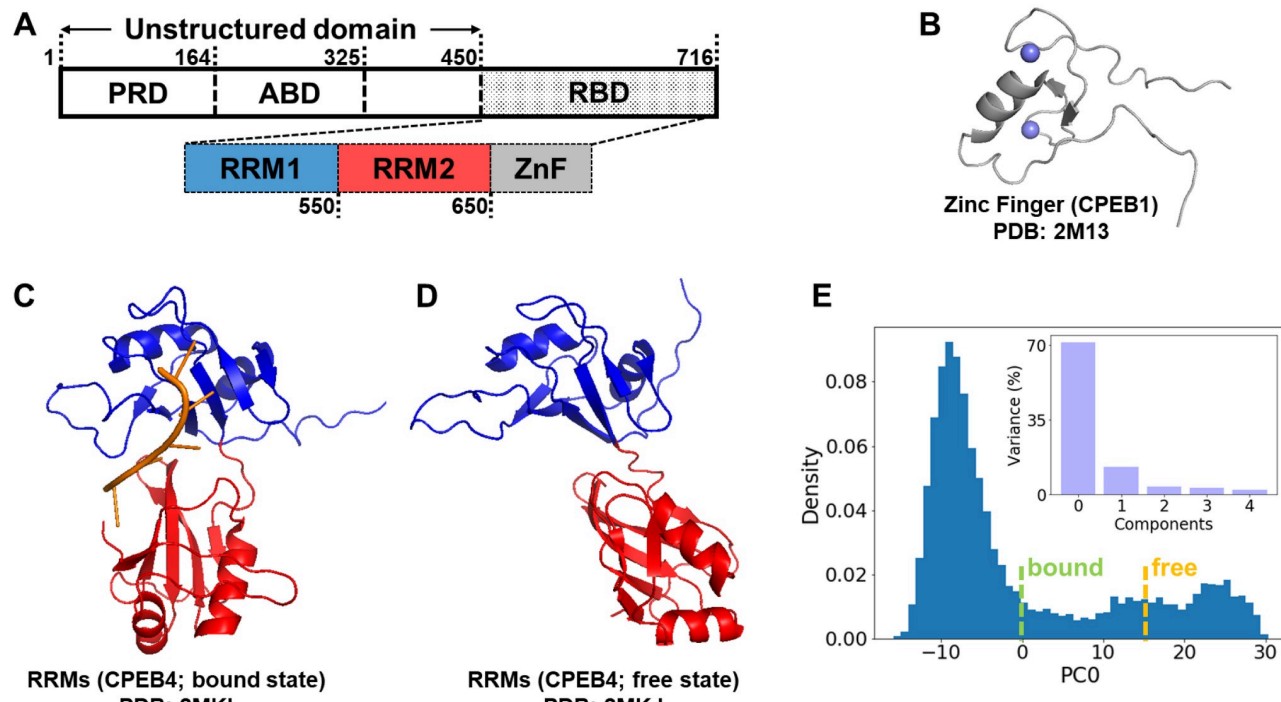

**Fig 2. A**. A diagram of the CPEB3 subdomains including a prion-like domain (PRD), an actin-binding domain (ABD), and an RNA-binding domain (RBD). The RBD consists of two RNA recognition motifs, RRM1 and RRM2 (colored in blue and red hereinafter), and one Zinc-finger domain (ZnF, colored in grey). **B**. The NMR structure of CPEB1 ZnF (Zinc ions are shown in purple). **C**. The NMR structure of CPEB4 RRMs binding with target mRNA (shown in orange). **D**. The NMR structure of free CPEB4 RRMs. **E**. The histogram for the distribution of first principal component, PC0, sampled during 20 equilibrium simulation trajectories of CPEB3 RRMs. Embedded: Variance contributed by the first five principal components from the principal component analysis. The PC0 values for the NMR structures of the RNA-bound state and the free state are shown by green and yellow dashed lines, respectively.

divided into two distinct subgroups based on the sequence similarity of the RBD: Taking the mouse CPEB family as an example, the RBDs of CPEB2–4 have 96% pairwise sequence identities, while the CPEB1 RBD has only 42% pairwise sequence identity with those of CPEB24, as shown in S1(A) Fig. The RBDs of CPEB3 homologs in other species, like CPEB3 in human, CPEB3 in rat, and Orb2 in Drosophila, have more than 86% pairwise sequence identity with the RBD of mouse CPEB3, as shown in S1(B) Fig. In this paper, we have studied the sequence of mouse CPEB3-RBD since a large fraction of the experimental work has been performed on mouse CPEB3.

As shown in Fig 2C, 2D and 2B, the NMR structures for the RRMs [23] and the ZnF [24] of some CPEB homologs have been solved. These were used as templates to build the corresponding structures for mouse CPEB3 via Modeller [25]. The tandem RNA recognition motifs are essential for sequence-specific recognition. They act by forming direct residue-base contacts with target mRNAs, while the zinc-finger domain only contributes to the RNA-binding affinity but not to the specificity of binding [26, 27]. The NMR structures of the RRMs suggest that there is a conformational change from the free state which is in an open conformation to the RNA-bound state which is closed. To study this closure motion, we first ran equilibrium simulations of the free RRMs using the AWSEM force field and conducted a cartesian principal component analysis for all the structures sampled during the equilibrium trajectories. Fig 2E maps the distribution of all frames onto the most significant principal component, PC0. The distribution displays two distinct modes for this principal component: one mode where PC0 ranges from -10 to 0 and another mode where PC0 ranges from 10 to 30. The NMR structure of the bound RRMs has a PC0 value near the first mode while the NMR structure of the free RRMs has a PC0 value near the second mode. We therefore use the PC0 value as an order parameter to distinguish between the open structures and the closed structures of the RRMs.

To model the full length RBD, we attached the ZnF domain to the C-terminal of the RRMs and ran AWSEM simulations to relax that initial structure. Clustering analysis for the final frames of sixty independent simulations suggests that the full-length RBD can adopt either the open conformation or the closed conformation, as shown in Fig 3A and 3B. The probability contact map for ZnF/RRM2 inter-domain contacts (Fig 3C) shows that the aromatic beta sheet surface (F669, F670, Y679 and Y680) on the ZnF consistently leads to an inter-domain interaction with RRM2. This interaction between the znic finger domain and the RNA recognition motif 2 was proposed previously based on the NMR data [24] and we see it emerges also in our model here.

## 2.2 Interaction with SUMO2 guides the RBD to select a conformation that favors RNA binding

Monomeric CPEB3 is SUMOylated in basal neurons. SUMO2, even after it has become covalently attached to CPEB3, can also non-covalently interact with the SUMO-Interacting Motif (SIM) in CPEB3. Classical SIMs have canonical consensus sequences which can interact with the second beta strand, $\beta$2, of SUMO2 to form an inter-molecular beta sheet [28]. Bioinformatic searches have identified one potential SIM at an exposed strand in the RNA recognition motif 1 (RRM1) of CPEB3. [16] The binding energy between this SIM and SUMO2 protein, calculated via the AWSEM Hamiltonian, is relatively large as compared with canonical SIM/SUMO2 complex, as shown in S2 Fig. Thus the interaction between SUMO2 and RRM1 via the SIM is favorable. It is intriguing that the zinc-finger domains from other proteins, such as the ubiquitin ligase HERC2 [29] and CBP (CREB-binding protein/p300) [30], have been reported to bind with SUMO proteins, suggesting that the zinc-finger domain contains a

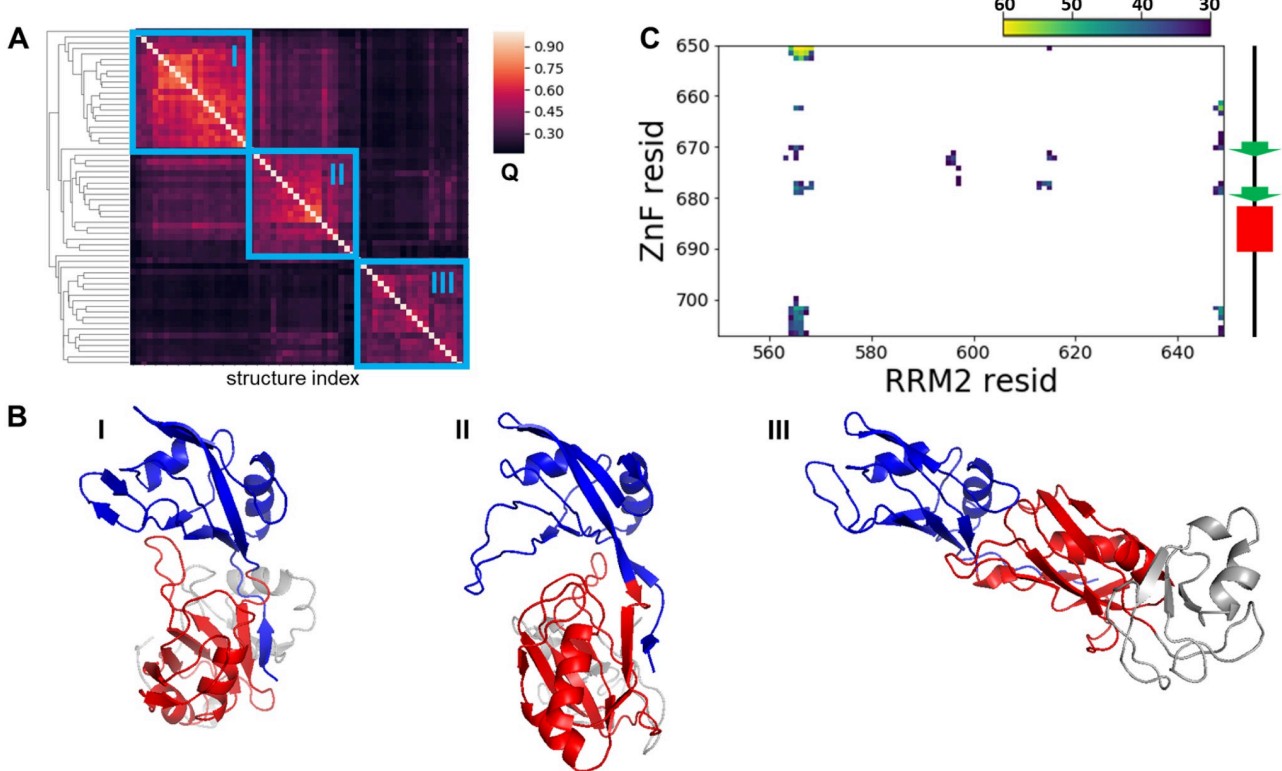

**Fig 3. A**. The 60 predicted structures of full length RBD that can be divided into 3 clusters using mutual Q (only including inter-domain residue pairs) as the structural similarity metric. **B**. Representative structures for the 3 clusters in Fig 3A. **C**. The frequency contact map for the inter-domain contact between the znic finger domain and the RNA recognition motif 2 (RRM2). Only contacts formed more than 30 times in 60 predictions are shown here. The diagram of the secondary structure of the ZnF domain is shown on the right side along the y-axis. Beta strands are colored in green and alpha helices are colored in red.

SUMO-interacting motif. The RBD of CPEB3, containing one canonical SIM and one zinc-finger domain, therefore may bivalently bind with SUMO2 specifically and stably.

To model the structure of the SUMO2/RBD complex, we first docked SUMO2 to the ZnF domain in the 60 structures of full-length RBD that were predicted and discussed in the last section while using the NMR structure of the SUMO1/CBP-ZnF complex (PDB ID: 2N1A) as a reference. We then added a weak biasing potential between the RRM1-SIM and the $\beta 2$ strand of SUMO2 to guide the formation of the inter-molecular beta sheet. The resulting structures were then allowed to relax without the use of the biasing potential. Final structures from these relaxation simulations were screened by calculating the inter-molecular structural similarity of RRM1-SIM/SUMO2 to canonical SIM/SUMO2 complex, as shown in S4 Fig. We calculated AWSEM energies and also counts of the number of minimally frustrated contacts [31] for selected structures as shown in Fig 4B. In a representative structure of the SUMO2/RBD complex (Fig 4C), SUMO2 forms two interfaces: one interface formed with the RRM1-SIM and the other formed with the ZnF domain.

To study the effects of the interaction with SUMO2 on the closure motion of RBD domain, we calculated the free energy profiles for the RBD domain by itself and for the SUMO2/RBD complex (Fig 5). For the RBD domain alone, there are two free energy basins: one for the open state (basin I in Fig 5A) and another for the closed state (basin II in Fig 5A). For the SUMO2/RBD complex in contrast, only the basin of closed state (basin III in Fig 5B) is found. The bivalent interaction with SUMO2 confines the conformation of CPEB3-RBD domain to the closed state.

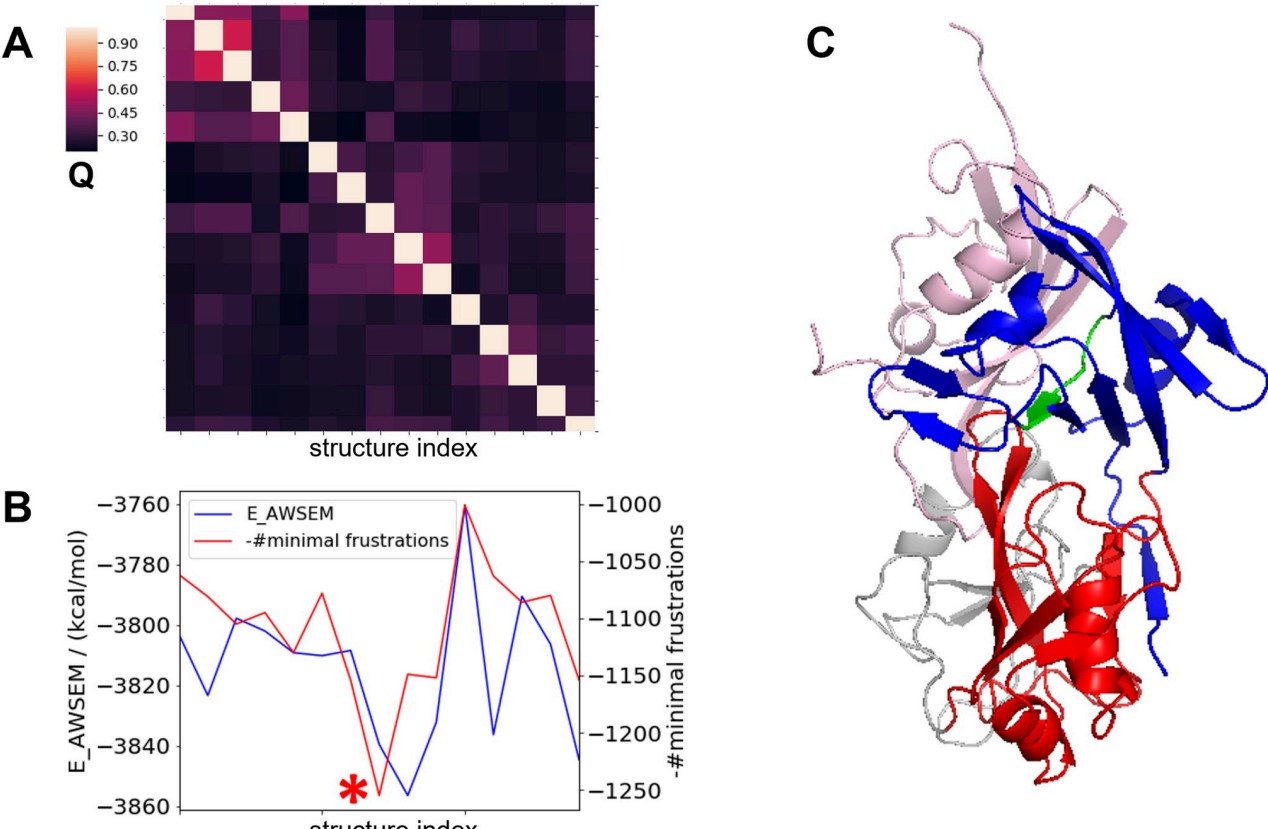

**Fig 4.** **A**. Clustering analysis for the predicted structures of the SUMO2/RBD complex using mutual Q (including inter-domain and inter-molecular residue pairs) as the structural similarity metric. **B**. Potential energy (blue line) and numbers of minimally frustrated contacts (red line) for the predicted SUMO2/RBD complex structures. The structure with the largest number of minimally frustrated contacts and having relatively low potential energy is selected as the representative structure (marked with a red star) and is shown in Fig 4C. **C**. The predicted structure of the SUMO2/RBD complex. The SUMO2 is colored in pink and the SIM in RRM1 is colored in green.

## 2.3 The SUMO2/RBD interaction increases the RNA binding affinity of the RBD

Our model shows that the specific binding of the RBD with SUMO2 makes the two RNA recognition motifs prefer the closed state, which is a favorable conformation for RNA binding. To further investigate the influence of this allosteric effect on the RNA-binding dynamics, we introduced a short piece of RNA containing 5 nucleotides to represent a target CPEB3 mRNA. We then used the AWSEM-3SPN2 force field along with an additional, sequence-specific protein-nucleic acid potential to simulate the combined protein/RNA system. The strength of the residue-nucleic acid potential for each base and residue pair was tuned by using the atomic contact numbers for each pair in the NMR structure of the RNA-bound RRMs as a reference. The NMR titration experiment shows that the RNA dissociation constant of RRMs is $15.8 \pm 6.6\mu M$ [23], which corresponds to an RNA-binding affinity of $6.4 \sim 6.9$ kcal/mol. In our simulations, we calibrated the overall strength of the residue-nucleic acid potential to obtain a similar RNA-binding affinity, 8 kcal/mol, for RRMs, as shown in Fig 6A.

The 1D free energy profiles for the protein/RNA systems suggest that the RNA-binding affinity of the SUMO2/RBD complex is around 2 kcal/mol higher than the affinity of the RBD alone for RNA. (Fig 6A). Upon projecting the free energy onto an additional order parameter,

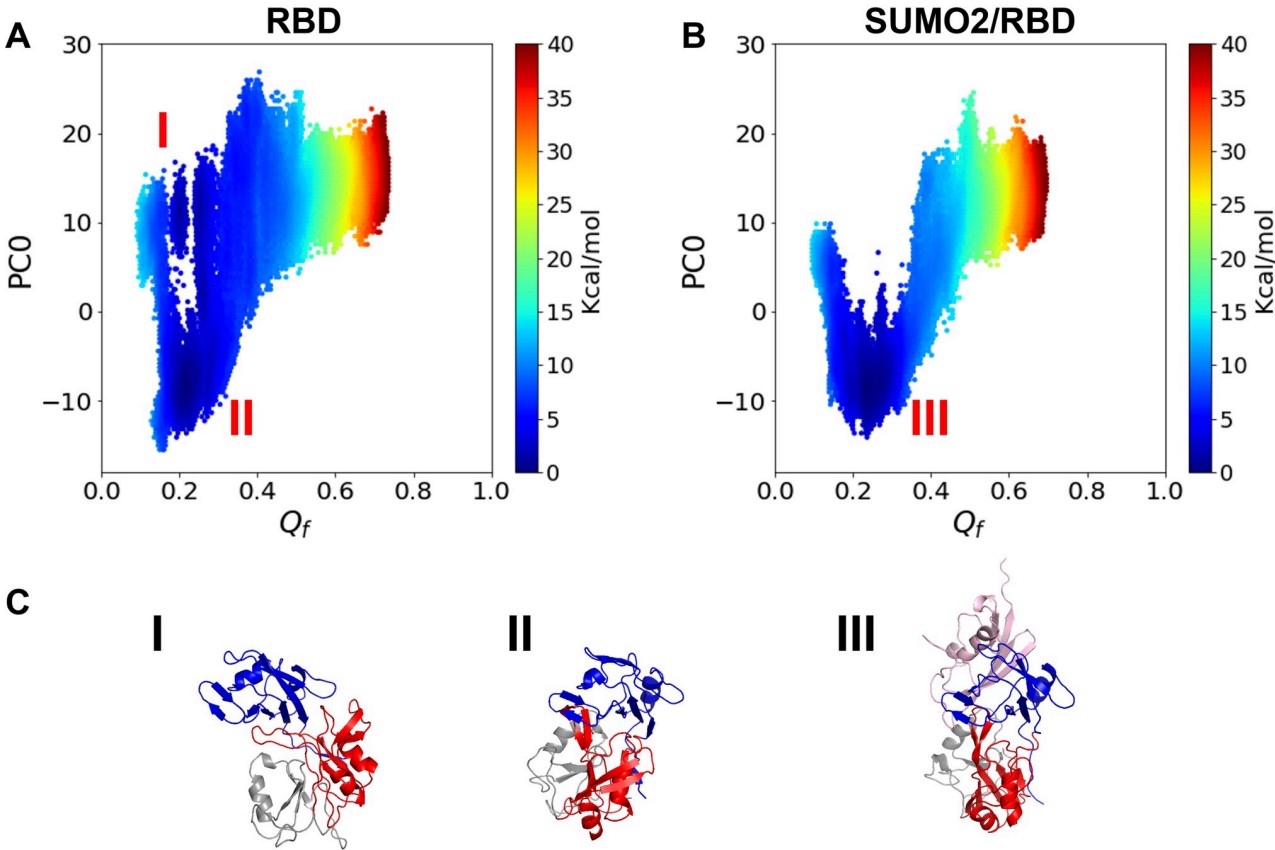

**Fig 5. A**. The free energy landscape for RBD plotted using $Q_f$ and PC0 as the two order parameters. $Q_f$ is the structural similarity metric to the NMR structure of free RRMs considering inter-domain residue pairs between RRM1 and RRM2 only. **B**. The free energy landscape for the SUMO2/RBD complex plotted using $Q_f$ and PC0 as the two order parameters. **C**. Representative structures for the three free energy basins in Fig 5A and 5B.

the principal component PC0, the 2D free energy profiles show a clear transition of dissociation pathway from one along path I to another along path II when SUMO2 becomes bound to the RBD (Fig 6B). Along path I (Fig 6C), the two RRMs open up during RNA dissociation, while along path II (Fig 6D) they remain closed because of the SUMO2/RBD interaction. As sketched in Fig 7, we propose that RNA dissociation from the RBD can be separated into two steps: first, RRM2 loses contact with the RNA and then rotates away from the binding pocket; following this, RRM1 dissociates from target RNA. For the SUMO2/RBD complex, however, it is difficult for RRM2 to dissociate from the RNA since RRM2 always stays in a ready-to-bind conformation when RRM1 is bound to RNA. RNA dissociation from the SUMO2/RBD complex therefore requires RRM1 and RRM2 to dissociate simultaneously from the RNA, thereby raising the barrier to dissociation. The existence of a subtle "shoulder" that appears around R = 1.3$nm$ in the 1D free energy profiles of the RRMs and RBD, but that is absent from that of SUMO2/RBD, supports this notion (Fig 6A). Due to the large unconstrained conformation space for RNA-free proteins, the entropy increase upon RNA dissociation in the RBD/RNA system is much larger than the entropy change in the SUMO2/RBD/RNA system. This larger entropy increase partially cancels the enthalpy increase during the RNA dissociation and results in a smaller free energy difference between the RNA-bound state and free state than would be seen without the difference in entropy. In conclusion, the binding of SUMO2 by RBD strengthens the RNA-binding affinity of the RBD domain by favoring the RNA-bound conformation of the tandem RNA recognition motifs.

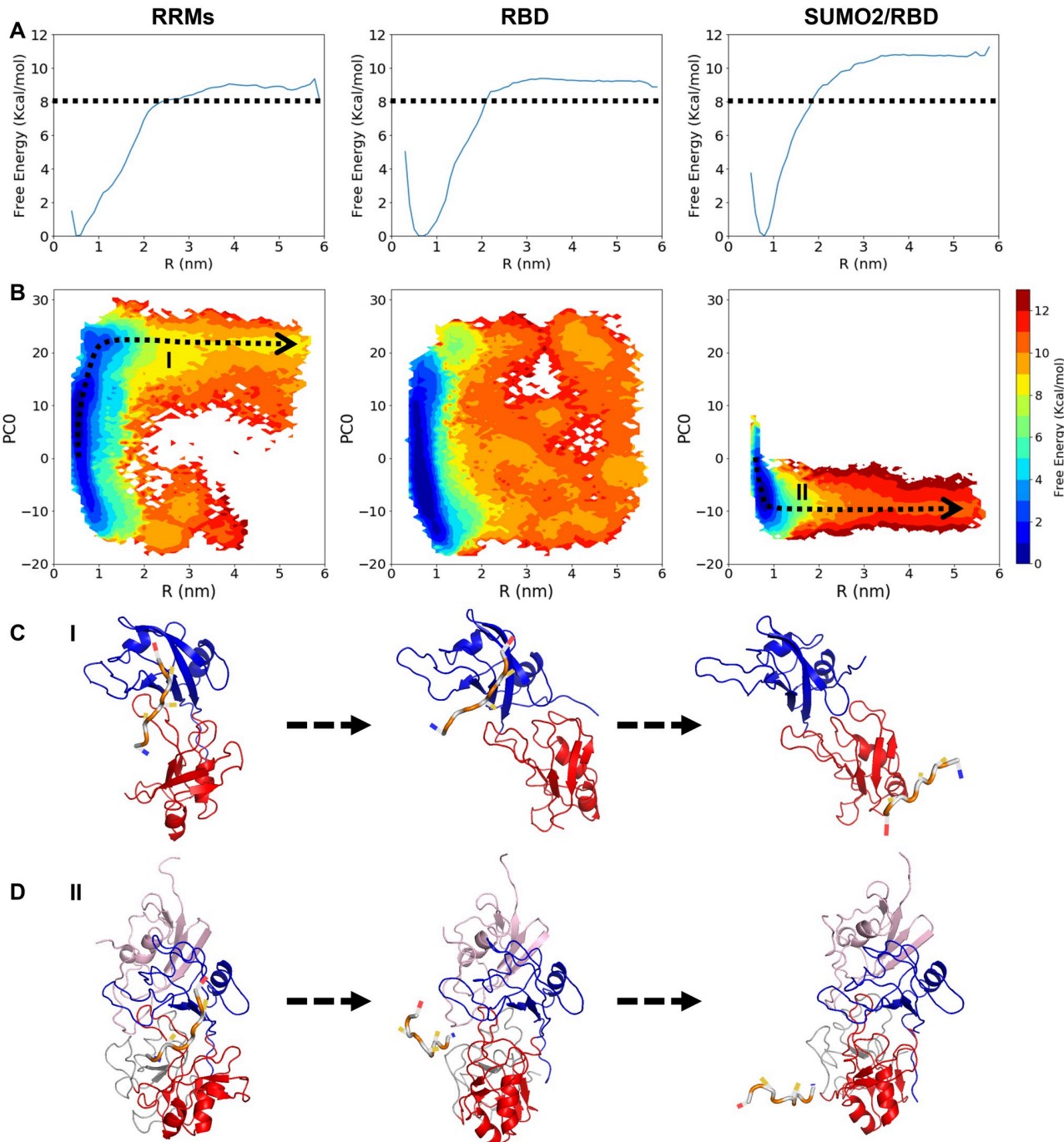

**Fig 6. A**. 1D free energy profiles for RNA dissociation from the RRMs (left), RBD (middle) and SUMO2/RBD complex (right), using R, the distance between RNA and the RNA binding pocket in RRMs, as the order parameter. The dashed black lines indicate a free energy difference of 8 kcal/mol from the RNA-bound state. **B**. 2D free energy landscapes for the RNA dissociation process from RRMs (left), RBD (middle) and SUMO2/RBD complex (right), using the principal component PC0 and R as two order parameters. **C**. Representative structures sampled from along the RNA dissociation pathway I from RRMs, as the dashed black arrow shown in Fig 6B. **D**. Representative structures sampled from along the RNA dissociation pathway II from SUMO2/RBD complex, as shown in Fig 6B. The main chain of the target RNA is colored in white and orange.

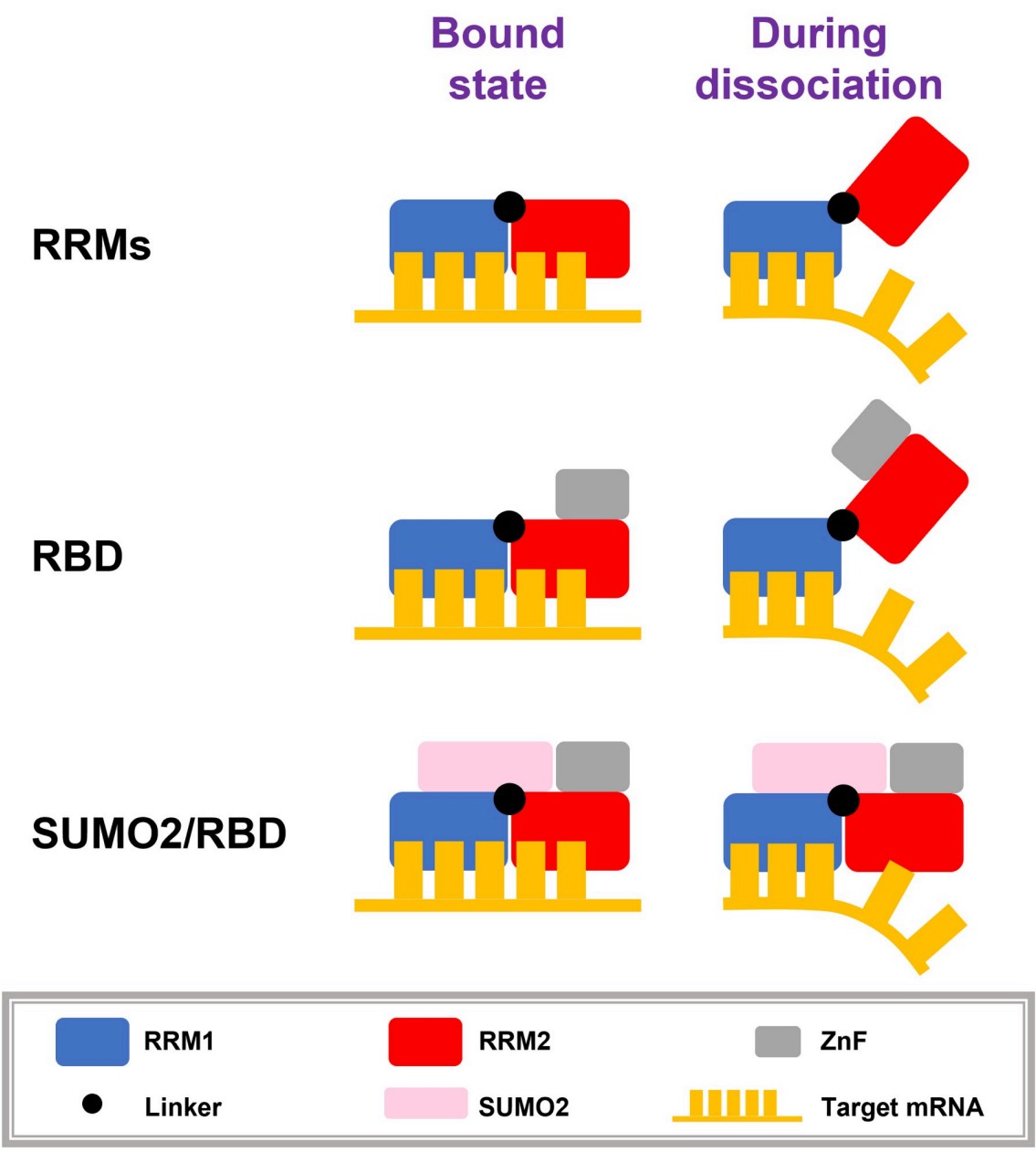

**Fig 7. Schematic structural diagrams of the RNA dissociation processes from RRMs, RBD, and SUMO2/RBD.** For the RRMs and the RBD, the RNA dissociation is a two-step process: RNA dissociation from RRM2 is followed by its dissociation from RRM1. For the SUMO2/RBD complex, RNA must dissociate from the two RRMs simultaneously.

## 3 Discussion

### 3.1 A negative feedback loop between CPEB3 and SUMO2 is completed by an equilibrium shift of RNA binding

The systems biology of SUMO2/CPEB3/mRNA suggests a negative feedback loop which addressed the potential problem of unregulated CPEB3 aggregation. SUMO2 mRNA, as a target of CPEB3, is activated by CPEB3 aggregates. While the translation products, SUMO2

proteins, can be attached to monomeric CPEB3 and prevent monomeric CPEB3 from further aggregation. CPEB3 aggregates which have already been formed still exist, however, after such a feedback loop. Therefore, the problem of unregulated translational enhancement by long-lasting CPEB3 aggregates still remains.

By combining notions from the systems biology with the effects of SUMO2 binding on regulating CPEB3's RNA-binding affinity, we see then that a more complete model of the switch of CPEB3's translational control can be sketched (Fig 8). Previous studies have proposed that the prion-like domain and the actin-binding domain of SUMOylated CPEB3 is buried based on bioinformatic searches for SUMOylation sites and SIMs in CPEB3. [16] After stimulation, CPEB3 is deSUMOylated and its PRD and ABD are exposed. Actin filaments are then able to bind with the exposed CPEB3-ABD and facilitate CPEB3 aggregation. CPEB3 fibers promote the translation of SUMO2 proteins, which are used for SUMOylating monomeric CPEB3 in synapses. SUMOylated CPEB3, having a higher RNA-binding affinity than a CPEB3 fiber, recruits target mRNAs from the CPEB3 fibers and sequesters them into P bodies. In this way, the synapses return to a new basal state after a stimulation pulse. The extent of the equilibrium shift of mRNA binding upon SUMOylation determines the efficacy of the switch in CPEB3's function in translational control. Free energy profiles in Fig 6A show that the difference of the RNA-binding affinity of the SUMO2/RBD complex and that of the RBD by itself is around 2 kcal/mol. Therefore, in equilibrium, the ratio of repressed target mRNA binding with SUMOylated CPEB3 to active target mRNA binding with deSUMOylated CPEB3 fiber would be around 30 in basal state (see detailed discussion in SI text). Accordingly, the function of CPEB3 could be sufficiently switched back to translational repression in basal synapses.

More experiments will be required to test the predictions of our model. Measuring the binding affinity of SUMO2 to CPEB3-RBD and solving the structure of the SUMO2/CPEB3-RBD complex would provide direct experimental tests of the calculations that we have performed. To assess whether the predicted SUMO2/RBD interaction is formed in full length CPEB3, the RNA-binding affinity of SUMOylated CPEB3 should be measured and compared with that of deSUMOylated CPEB3. Structural studies of monomeric CPEB3 and aggregated CPEB3 will be crucial to uncovering the functions of CPEB3 in long-term memory.

## 4 Methods

### 4.1 Definition of the structural similarity metric Q value

As a metric of structural similarity for 2 configurations, the Q value ranges from 0 to 1, lower values reflecting low similarity while Q = 1 indicates the structures are nearly exactly the same. The definition of the mutual Q for two structure A, B is shown below,

$$Q = \frac{1}{N}\sum_{ij} e^{-\frac{(r_{ij}^A - r_{ij}^B)^2}{2\delta_{ij}^2}}$$

where N is the total number of residue pairs considered in Q calculation, i, j are residue indices and $r_{ij}^A$ and $r_{ij}^B$ are the distances between residue i and residue j in structures A, B respectively. For intra-molecular residue pairs, $\delta_{ij} = |i - j|^{0.15}$, while for inter-molecular residue pairs, $\delta_{ij}^2$ is set to a constant value, 5 $A^2$. We included all the inter-molecular and inter-domain residue pairs for calculating mutual Q in clustering analysis and $Q_f$ in free energy analysis. For the Q values of RRM1-SIM/SUMO2 when comparing with the canonical SIM/SUMO2 template, we only considered inter-molecular residue pairs which form contacts in the template structure

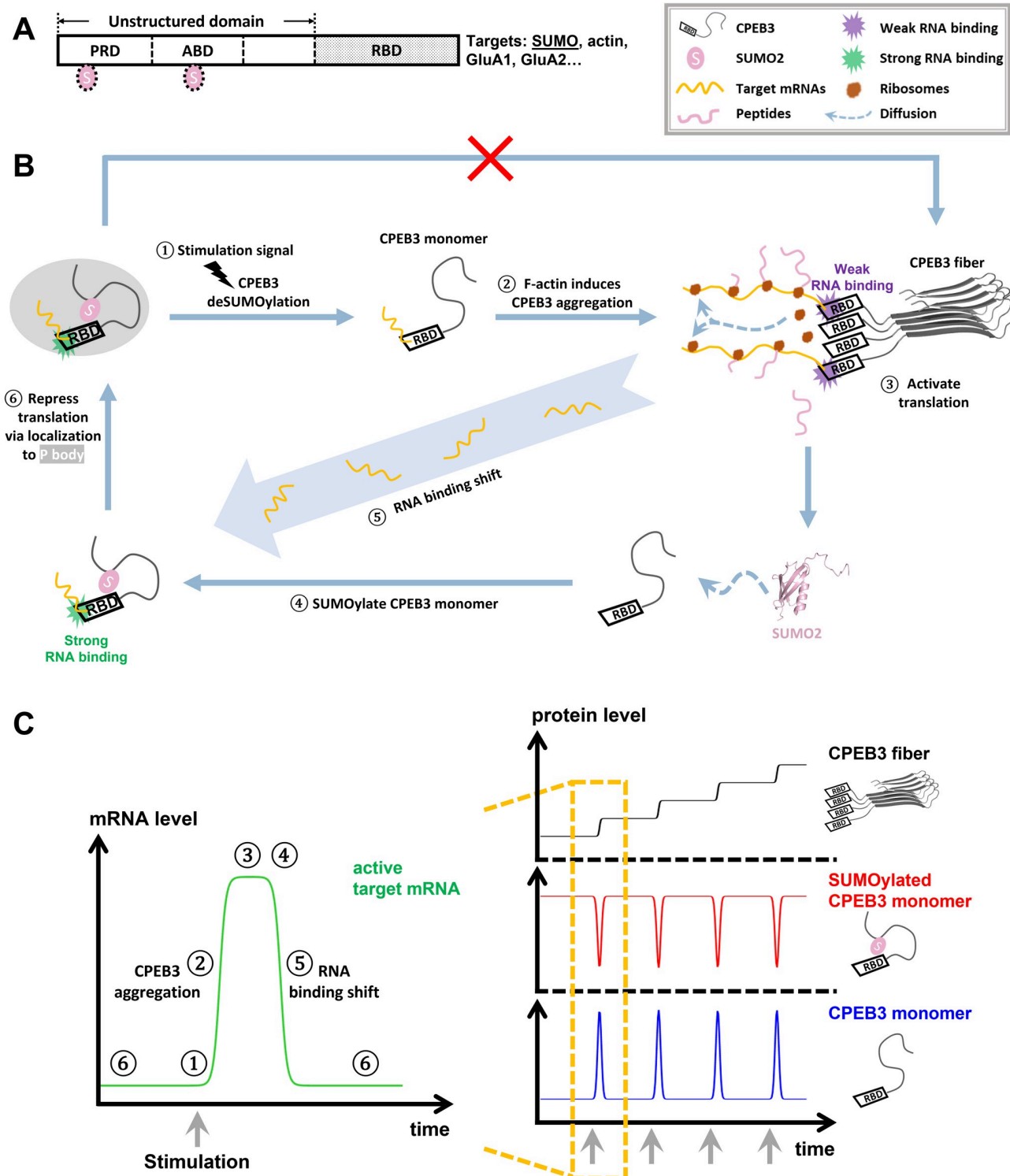

**Fig 8. A**. The two potential SUMOylated sites on CPEB3 are highlighted using two SUMO2 labels (pink). **B**. The complete negative feedback loop between CPEB3 and SUMO2: ① An external signal stimulates the synapses and triggers the deSUMOylation of basal CPEB3. ② CPEB3 exposes its prion-like domain (PRD) and actin-binding domain (ABD) and aggregates upon binding with actin filaments (F-actin). ③ CPEB3 fibers form local translation factory assembly lines to activate the translation of its target mRNAs, which includes SUMO2 mRNA. ④ Newly synthesized SUMO2 proteins are then used for the SUMOylation of monomeric CPEB3. ⑤ A shift of mRNAs from binding with CPEB3 fibers to binding with SUMOylated CPEB3 occurs due to the RNA-binding affinity difference. ⑥ SUMOylated CPEB3 binds with target mRNAs and recruits them into P bodies for translational repression. A legend is

shown at the upper-right corner. **C**. Sketches of the CPEB3 regulation mechanism for target mRNA activity in synapses. The active mRNA level during one feedback cycle, in response to one synaptic stimulation pulse, is illustrated in the left panel. The protein levels of different CPEB3 states in response to several stimulation pulses are sketched in the right panel. Following each stimulations pulse, synapses return to a new basal state where the CPEB3 fiber level has been risen. We see that a sequence of stimulation events acts as a ratchet, increasing the basal level of CPEB3 fibers in steps. Stimulation steps are labeled by grey arrows.

or in the predicted structure. In this work, we chose a distance threshold of 12 A to define contacts between residues.

## Supporting information

**S1 Text. Introduction to the AWSEM-3SPN2 force field.** Binding energy for SIM/SUMO2 complex. Additional potentials. Simulation methods: Equilibrum simulation of free RRMs; Equilibrum simulation of full length RBD; Structural prediction to the SUMO2/RBD complex; Free energy profiles for RNA dissociation; Free energy profiles for the closure motion of RRMs. Efficiency of the shift of RNA-binding equilibration.
(PDF)

**S1 Fig. Multiple sequence alignment for the RNA-binding domain (RBD) of CPEB homologs. A**. Four isoforms of mouse CPEB family. **B**. Homologs of mouse CPEB3, including CPEB3 in human, D4AD99 in rat, Orb2 in Drosophila and ApCPEB in Aplysia. For each residue, higher pairwise identity is marked by deeper blue color. Percentages of overall sequence identity are listed at the end of the alignment, comparing with mouse CPEB3.
(EPS)

**S2 Fig. Binding energy of canonical SIM/SUMO2 complex structure when scanning the sequence of full length CPEB3 over the SIM peptide.** The x axis is the residue index of the first amino acid in each CPEB3 peptide. The black line shows the binding energy for the original SIM peptide in the PDB structure (ID: 6JXW). The two green shaded regions highlighted peptides containing the predicted SIMs in CPEB3 via bioinformatic search (V273-V283; P484-W489).
(EPS)

**S3 Fig. The 20 predicted structures of free RRMs can be divided into 3 clusters using mutual Q (only including inter-domain residue pairs) as the structural similarity metric.** Representative structures for these 3 clusters are shown on the right.
(EPS)

**S4 Fig. A**. One example of the initial structure for the SUMO2/RBD complex structure prediction. The SUMO2 protein (pink) is docked into the ZnF domain based on PDB structure 2N1A. $\beta$2 strand of SUMO2 is colored in orange and the RRM1-SIM is colored in green. **B**. Canonical SIM/SUMO2 complex (PDBID: 6JXW). SUMO2 is colored in pink. The SIM peptide is colored in blue, and the green fragment inbetween corresponds to the RRM1-SIM. The structure of SUMO2 and the green fragment is used as the reference to calculate the $Q_c$ value of SUMO2/RRM1-SIM in predicted SUMO2/RBD complex. **C**. The distribution of $Q_c$ value of SUMO2/RRM1-SIM in 60 predicted SUMO2/RBD complex structures. Structures with $Q_c$ larger than 0.2 is selected for further evaluation.
(EPS)

**S5 Fig. A**. The free energy landscape for RBD/RNA using $Q_f$ and PC0 as the two order parameters. **B**. The free energy landscape for the SUMO2/RBD/RNA complex using $Q_f$ and PC0 as the two order parameters. **C**. Representative structures for the three free energy basins in

Figure A and B.
(EPS)

**S6 Fig.  A**. The 1D Free energy profile for RNA-free RBD and SUMO2/RBD complex using PC0 as the order parameter. **B**. The 1D Free energy profile for RNA-bound RBD and SUMO2/RBD complex using PC0 as the order parameter. The free energy of the open state (PC0 equals $10 \sim 30$) is significantly increased upon RBD binding with SUMO2 protein.
(EPS)

**S7 Fig. The two RNA dissociation pathways from RBD.** These correspond with the pathway I and pathway II shown in Fig 5B. The starting point of the dissociation pathways is marked by a white star.
(EPS)

**S1 Movie. RNA dissociation pathway I from RRMs shown in Fig 5B.**
(MP4)

**S2 Movie. RNA dissociation pathway II from SUMO2/RBD complex shown in Fig 5B.**
(MP4)

**S3 Movie. RNA dissociation pathway I from RBD shown in S7 Fig.**
(MP4)

**S4 Movie. RNA dissociation pathway II from RBD shown in S7 Fig.**
(MP4)

## Author Contributions

**Conceptualization:** Xinyu Gu, Nicholas P. Schafer, Peter G. Wolynes.

**Data curation:** Xinyu Gu.

**Formal analysis:** Xinyu Gu.

**Funding acquisition:** Peter G. Wolynes.

**Investigation:** Xinyu Gu, Nicholas P. Schafer.

**Methodology:** Xinyu Gu, Nicholas P. Schafer, Carlos Bueno, Wei Lu, Peter G. Wolynes.

**Project administration:** Peter G. Wolynes.

**Resources:** Peter G. Wolynes.

**Supervision:** Peter G. Wolynes.

**Validation:** Xinyu Gu, Nicholas P. Schafer, Peter G. Wolynes.

**Visualization:** Xinyu Gu, Peter G. Wolynes.

**Writing – original draft:** Xinyu Gu, Peter G. Wolynes.

**Writing – review & editing:** Xinyu Gu, Nicholas P. Schafer, Carlos Bueno, Wei Lu, Peter G. Wolynes.

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
