## [Decision Letter · Decision Letter 0]

1 Sep 2022

Dear Peter,

Thank you very much for submitting your manuscript "A structural dynamics model for how CPEB3 binding to SUMO2 can regulate translational control in dendritic spines" for consideration at PLOS Computational Biology. As with all papers reviewed by the journal, your manuscript was reviewed by members of the editorial board and by several independent reviewers. The reviewers appreciated the attention to an important topic. Based on the reviews, we are likely to accept this manuscript for publication, providing that you modify the manuscript according to the review recommendations.

Sincerely,

Shi-Jie Chen

Academic Editor

PLOS Computational Biology

Nir Ben-Tal

Section Editor

PLOS Computational Biology

[LINK]

Reviewer's Responses to Questions

**Comments to the Authors:**

Reviewer #1: Gu and others built the models of RNA binding to CPEB3-RBD with and without the protein SUMO2. The results of coarse-grained AWSEM molecular dynamic simulations suggest the influence of SUMO2 on the RNA affinity as well as the dissociation pathway. Both the system and the designation of the simulations are interesting. However, I have a few comments about this manuscript.

The authors state that they have calculated the free energy profile for RNA dissociation from the SUMO2/CPEB3-RBD complex. These calculations show that the RNA-binding free energy for the SUMO2/CPEB3-RBD complex is 2 kcal/mol larger than that for isolated RBD in deSUMOylated CPEB3 (Ref 22). Could the authors describe the difference between the systems in the two papers? It seems that these systems are similar.

In this manuscript, the RBD structure is constructed via Modeller. In addition, the SUMO2/RBD complex is obtained by docking SUMO2 to the ZnF domain in the 60 structures of full-length RBD. In my opinion, the authors should provide the reliability of the MD model. Maybe there are no experimental affinity data of RNA to SUMO2/RBD. But I think some key interactions, binding poses, or some structural characteristics can be used to prove the MD simulation results.

In section 2.2, the authors claim that “SUMO2, even after it has become covalently attached to CPEB3, can also non-covalently interact with the SUMO-Interacting Motif (SIM) in CPEB3.” Could the authors state that in general SUMO2 is covalently or non-covalently attached to CPEB3?

Reviewer #2: The manuscript presents a computational study on the assembly of the CPEB3 protein to sumo and RNA. This is a complex and rich system due to its large size and high flexibility. It is much more complex than typical bimolecular reactions of protein-protein or protein-DNA. Its biological relevance is high. The manuscript is nicely written and the results are clearly presented.

Comments

1. The study involves MD simulations using a simple model on a very complex system. Some calibration of the model is performed. This is unique and appreciated. The discussion on the experimental interpretation is not clear enough and should be better communicated. It is hard to tell if the manuscript is addressed to a computational community of modeling or to the biological community who is interested in CPEB3. It will be better if it will be addressed to both communities. For example, the last section seems to be very different than the rest of the manuscript. The integration between the different parts may improve the clarity of the manuscript.

2. For example, a schematic figure that supports the intro may assist in clarifying the details. Perhaps some parts of the last figure can be moved to an introductory figure.

3. A “conclusions” section is missing. Also the Discussion part includes only a single subsection? Was it supposed to include an additional part? Is the submitted manuscript complete?

4. How was the RNA modeled? Is there anything that differentiate the RNA modeling from DNA models?

5. Why calibrating the affinity was done to 8kcal.mol while the exp value is closer to 6 kcal/mol?

6. A sentence that explains what Sumo2 is relative to sumo, might be helpful

Reviewer #3: This manuscript is part of an ongoing effort from Dr. Wolynes and colleagues to provide insights into an important problem via modelling: what are the consequences of conversion of CPEB proteins from monomer to aggregates in the synapse and implications in stabilization of memory. In this manuscript, through modelling, they attempted to address how translation activation by aggregated CPEB3 protein can be prevented from running away. This is important since the aggregates are supposed to be self-sustaining. The model takes into consideration three factors: CPEB3 binds mRNA, SUMOlyation prevents CPEB3 aggregation and promotes p-body formation, and aggregated CPEB3 translates SUMO2. The major conclusion from the modelling is that the binding of SUMOlyated monomeric CPEB3 to mRNA is stronger than that of deSUMOylated aggregated CPEB3. The model suggests aggregated CPEB3, by translating SUMO2, can restrict its own activity. Controlling persistent translation activation by translating a modifier although not entirely novel, nonetheless an important insight. However, the authors should consider and discuss following issues:

1) What is the relationship between stronger mRNA binding and P-body formation and/or translation inactivation? If SUMOlyation is the major driver for p-body formation and P-body prevents translation, then what is the contribution of stronger mRNA binding. It is also unclear why a weaker mRNA binding would promote translation. In fact, stronger mRNA binding can promote translation.

2) The model is based on published experimental observations. However, all published studies of CPEB3 utilizes in-cell translation to understand the relationship between CPEB3-SUMolytaion, aggregation and translation. Since several of these experiments were done in the background of endogenous unmodified CPEB3 in presence of other regulators of translation and aggregation, it remains unclear whether small fraction of SUMOlyated CPEB3 or bulk-SUMOlyation is responsible for the translation and aggregation differences. These two scenarios would invoke different models.

3) Finally, the aggregation of CPEB3 in a specific synapse supposedly has two consequences: transition from repression to activation and maintaining an altered state of the synapse. The model shows aggregation provides a transient burst of activation and SUMOlyation returns the process to the initial state of the synapse. This is compatible with the transition from repression to activation, but it does not accommodate the second consequence. If it returns to the original basal state, then aggregation-mediated translation would not contribute the maintenance of the modified state of the synapse. Or am I missing something important here? Anyway, this needs to be clarified in the discussion.

**Have the authors made all data and (if applicable) computational code underlying the findings in their manuscript fully available?**

Reviewer #1: Yes

Reviewer #2: Yes

Reviewer #3: Yes

PLOS authors have the option to publish the peer review history of their article (what does this mean?). If published, this will include your full peer review and any attached files.

Reviewer #1: No

Reviewer #2: No

Reviewer #3: No

Figure Files:

Data Requirements:

Reproducibility:

References:

---

## [Decision Letter · Decision Letter 1]

14 Oct 2022

Dear Peter,

We are pleased to inform you that your manuscript 'A structural dynamics model for how CPEB3 binding to SUMO2 can regulate translational control in dendritic spines' has been provisionally accepted for publication in PLOS Computational Biology.

Best regards,

Shi-Jie Chen

Academic Editor

PLOS Computational Biology

Nir Ben-Tal

Section Editor

PLOS Computational Biology

Reviewer's Responses to Questions

**Comments to the Authors:**

Reviewer #1: The authors have addressed my questions. I believe it is ready to be accepted.

Reviewer #2: My concerns have been addressed in the revised manuscript

Reviewer #3: In the revised version of the manuscript the authors have adequately addressed the main issues.

**Have the authors made all data and (if applicable) computational code underlying the findings in their manuscript fully available?**

Reviewer #1: Yes

Reviewer #2: Yes

Reviewer #3: None

PLOS authors have the option to publish the peer review history of their article (what does this mean?). If published, this will include your full peer review and any attached files.

Reviewer #1: No

Reviewer #2: No

Reviewer #3: No

---

## [Editor Report · Acceptance letter]

3 Nov 2022

PCOMPBIOL-D-22-01067R1 

A structural dynamics model for how CPEB3 binding to SUMO2 can regulate translational control in dendritic spines

Dear Dr Wolynes,

I am pleased to inform you that your manuscript has been formally accepted for publication in PLOS Computational Biology. Your manuscript is now with our production department and you will be notified of the publication date in due course.

With kind regards,

Zsofia Freund
